Enhancement of a protocol purifying T1 lipase through molecular approach

Che Hussian Che Haznie Ayu 1 2 3
Raja Abd. Rahman Raja Noor Zaliha 1 2 3 rnzaliha@upm.edu.my
http://orcid.org/0000-0002-0280-7735 Thean Chor Adam Leow 1 2 4
Salleh Abu Bakar 2 5
Mohamad Ali Mohd Shukuri 2 5
1 Institute of Bioscience, Universiti Putra Malaysia , Serdang, Selangor , Malaysia
2 Enzyme and Microbial Technology Research Center, Universiti Putra Malaysia , Serdang, Selangor , Malaysia
3 Department of Microbiology, Faculty of Biotechnology and Biomolecular Sciences, Universiti Putra Malaysia , Serdang, Selangor , Malaysia
4 Department of Cell and Molecular Biology, Faculty of Biotechnology and Biomolecular Sciences, Universiti Putra Malaysia , Serdang, Selangor , Malaysia
5 Department of Biochemistry, Faculty of Biotechnology and Biomolecular Sciences, Universiti Putra Malaysia , Serdang, Selangor , Malaysia
Uversky Vladimir
Electronic publication date: 2018 Nov 16
Publication date: 2018
Volume: 6
Electronic Location ID: e5833
Received 2018 Aug 8; Accepted 2018 Sep 27
Copyright: © 2018 Che Hussian et al.
Copyright year: 2018
Copyright holder: Che Hussian et al.
License: This is an open access article distributed under the terms of the Creative Commons Attribution License, which permits unrestricted use, distribution, reproduction and adaptation in any medium and for any purpose provided that it is properly attributed. For attribution, the original author(s), title, publication source (PeerJ) and either DOI or URL of the article must be cited.
License URL: https://creativecommons.org/licenses/by/4.0/

Keywords: Isoelectric point, E. coli, Lipase, Purifications, Site-directed mutagenesis

Funding: The Ministry of Science, Technology and Innovation Malaysia (MOSTI) under Malaysia Technology Development Centre (MTDC) for the financial aid through CRDF Bridging Fund Vot number 6364703 This work was supported by the Ministry of Science, Technology and Innovation Malaysia (MOSTI) under Malaysia Technology Development Centre (MTDC) for the financial aid through CRDF Bridging Fund (Vot number 6364703). The funders had no role in study design, data collection and analysis, decision to publish, or preparation of the manuscript.

==============================
T1 Lipase is a thermostable secretary protein of Geobacillus zalihae strain previously expressed in a prokaryotic system and purified using three-step purification: affinity 1, affinity 2, and ion exchange chromatography (IEX). This approach is time consuming and offers low purity and recovery yield. In order to enhance the purification strategy of T1 lipase, affinity 2 was removed so that after affinity 1, the cleaved Glutathione S-transferase (GST) and matured T1 lipase could be directly separated through IEX. Therefore, a rational design of GST isoelectric point (pI) was implemented by prediction using ExPASy software in order to enhance the differences of pI values between GST and matured T1 lipase. Site-directed mutagenesis at two locations flanking the downstream region of GST sequences (H215R and G213R) was successfully performed. Double point mutations changed the charge on GST from 6.10 to 6.53. The purified lipase from the new construct GST tag mutant-T1 was successfully purified using two steps of purification with 6,849 U/mg of lipase specific activity, 33% yield, and a 44-fold increase in purification. Hence, the increment of the pI values in the GST tag fusion T1 lipase resulted in a successful direct separation through IEX and lead to successful purification.

Introduction

Lipases or acylglycerol hydrolases (E.C. 3.1.1.3) are enzymes that catalyze the hydrolysis of long chain triglycerides with the formation of diacylglycerides, monoglycerides, glycerols, and free fatty acids at a lipid-water interface (Fickers et al., 2006). Lipases are considered prime players in most biotechnological application, as the flexibility of their functional properties and unique inherent specificity and stability have no doubt attracted attention as biocatalysts of the future (Hasan, Shah & Hameed, 2006). Thermostable lipases are in high demand for industrial applications because of their high conversion rate and stability at high temperature. They have huge potential in commercial applications, such as in the production of food additives (for flavor modification), fine chemicals (synthesis of esters), detergents (hydrolysis of fats), wastewater treatment chemicals (decomposition and removal of oil substances), cosmetics (removal of lipids), pharmaceuticals (digestion of oil and fats in foods), leather (removal of lipids from animals skins), and medical products (blood triglycerides assay) (Gupta, Gupta & Rathi, 2004; Gupta et al., 2005; Kamini et al., 2000; Pandey et al., 1999). A review of several findings for thermostable lipases has shown setbacks in obtaining lipase in a large scale due to the cost-intensive demand as compared to other commercial enzymes. Ideally, to fully utilize the beneficial properties of thermostable lipase, the overall production process and its commercialization should be less expensive and less time consuming. Consequently, the homogeneity and purity of these enzymes is fundamental to their overall applications as biocatalysts in industrial enterprises. Basically, most regular lipase purification strategies require at least three to five step methods (Brusw & Gerasimov, 1977; Gupta et al., 2005; Saxena et al., 2003). However, nowadays researchers have been working on molecular engineering as a toolbox to facilitate protein purification (Dheeman, Gary & Frias, 2011; Sassenfeld, 1990). Since the successful application of these enzymes requires their production on a large scale, commercializing them will require a novel and simplified approach toward their large-scale production and purification aimed at reducing cost of enzyme production. Therefore, a more efficient purification strategy must be established for future commercialization.

One novel lipase that has high potential to be marketed is T1 lipase, a thermoalkaliphilic enzyme secreted by Geobacillus zalihae strain T1. T1 lipase was previously overexpressed in pGEX4T1 vector in prokaryotic system, Escherichia coli BL21(DE3)pLysS as Glutathione S-transferase (GST) tagged fusion T1 lipase and be captured at first step through an affinity chromatography. However, a problem was determined in the next step of separation when the digested T1 lipase and GST tag could be separated directly through ion exchange chromatography (IEX) as the isoelectric point (pI) value were too close. Thus, another affinity step known as affinity 2 was used to captured digested GST tag while eluting lipase at flow through (Leow et al., 2007). Nevertheless, there is still some GST tag eluting together with T1 lipase. To further achieve higher purity of T1 lipase, an additional step of anion exchange chromatography was reported with a substantial enzyme recovery and purity for crystallization (Aris et al., 2014).

The purification of lipase is not an easy task due to the complexity of proteins and their biological environment; therefore, separation and purification often account for a major proportion of commercialization and production costs. One useful purification technique used to separate proteins based on protein charges is IEX, an outstanding technique in protein purification either in the intermediate stage or during the polishing step. The pI of target proteins is initially determined followed by identification of suitable matrix either anion or cation exchanger. In this paper, the overall workflow for T1 lipase purification was shown in Fig. 1. The enhancement of a protocol aiming at purifying a thermostable T1 lipase was developed in two steps: (i) affinity chromatography to purify the lipase fused to GST tag out of E. coli lysate; and (ii) anion exchange chromatography to finally separate lipase and GST-T1 lipase. As the pI of GST and T1 lipase are close, performing anion chromatography directly after first affinity chromatography is initially not possible. To circumvent this issue, site-directed mutagenesis on two residues was conducted to generate a GST tag mutant (GSTm) with a higher pI. The resulting GSTm-tag could be used for the first affinity chromatography, then separated from T1 lipase by anion exchange chromatography.

Figure 1 Overall workflow of purification steps of GSTm-T1 lipase.

Materials and Methods

Bacterial strains and plasmids

Escherichia coli Top 10 and BL21(DE3)pLysS cells harboring the recombinant pGEX/T1 of G. zalihae strain T1 (AY260764.2) were used for cloning and expression, respectively. The recombinant pGEX/T1 was used as a source for rational design.

Isoelectric point value computation

The computation of pI value of GST tag protein and mature T1 lipase was performed using ExPASy-compute pI/MW at http://www.expasy.ch/tools/pI-tool.html. Two point mutations H215R and G213R were conducted at the C-terminal end of the GST tag and the theoretical pI value computed using the above software.

Molecular modification by site-directed mutagenesis

Mutagenic primers (forward and reverse) were designed by referring to the point of mutation (bold and underlined) at C-terminal end of GST tag sequence. The designed mutagenic primers are, MutGST Forward: 5′-GCC ACG TTT GGT GGT AGA GAC AGA CCT CCA AAA TCG GAT-3′ and MutGST Reverse: 5′-ATC CGA TTT TGG AGG TCT GTC TCT ACC ACC AAA CGT GGC-3′. Site-directed mutagenesis was performed using a Stratagene QuikChange® Lightning kit according to manufacturer instructions. The recombinant mutagenic plasmid of positive transformant was confirmed by DNA sequencing before transformation into expression host E. coli BL21(DE3)pLysS.

Transformation into expression system

Once the sequencing results showed that the amino acids were successfully substituted, five μL of mutated vector was transferred into competent cells E. coli BL21(DE3)pLySs (20 μL) and incubated for 30 min on ice. The samples in the Eppendorf tubes were pulse heated in water bath at 42 °C for 90 s and immediately incubated at 37 °C for 1 h. Then, the sample was spread on LB-Tributyrin agar containing ampicillin and chloramphenicol antibiotic and incubated at 37 °C for 1 day. Transformed colonies were screened on Triolein and Rhodamine agar plates.

Expression of GSTm-T1 lipase (mutated)

Mutagenic fusion lipase (GSTm-T1) expression was conducted in 200 mL LB containing ampicillin (50 mg/mL) and chloramphenicol (35 mg/mL) at 37 °C with an agitation rate of 200 rpm. The culture was induced with 0.025 mM of Isopropyl ß-d-thiogalactopyranoside for 12 h and harvested at 10,000 rpm and 4 °C for 10 min and pellet stored in a freezer at −80 °C.

Cell lysis and clarification of lysate

The pellets were resuspended with 20 mL of phosphate buffered saline (PBS) pH 8.0 containing five mM DTT and sonicated for 4 min (30 s on, 30 s off). Cell debris was removed by centrifugation at 12,000×g for 30 min. Pelleted cell debris was discarded and supernatant was taken as crude enzyme and filtered by using a 0.45 μm syringe filter.

Purification

Purifications of GSTm-T1 lipase and GST-T1 lipase (wild type) conducted as a comparison study. Crude enzyme (20 mL) was subjected to affinity chromatography column XK16/20 containing five mL of Glutathione Sepharose Fast Flow from Amersham Bioscience. The 20% ethanol in resin firstly was removed by distilled water with a flow rate of one mL/min for 10 column volume (CV). Then, resin was equilibrated with 3–5 CV of PBS pH 7.4. The equilibrated column was attached to fast-performance liquid chromatography (FPLC) system (AKTA Explorer, GE Healthcare, Chicago, IL, USA). Crude enzyme was loaded onto the column and operated by AKTA Explorer system at a flow rate of one mL/min. Fractions were collected and fusion lipases eluted with thrombin cleavage buffer pH8.0 (50 mM Tris–HCl), supplemented with 10 mM reduced glutathione.

Fusion lipase fractions from affinity chromatography were pooled and absorbance at 280 nm was read. Then, bovine thrombin was added into the pooled samples and incubated at 20 °C for 20 h prior to the cleavage of GST and matured T1 lipase. The incubated samples were analyzed by SDS–PAGE to make sure GST and matured T1 lipase were cleaved and separated. The sample was then subjected to buffer exchange by using Sephadex G25. The digested fusion lipase was separated by injecting into Q sepharose FF with a flow rate of 0.5 mL/min. After the sample injection, the flow-through fraction (5 CV) was collected using binding buffer as the eluent, followed by washing unbound proteins (5 CV) and eluted using salt gradient (20 CV) toward 25 mM Tris–HCl buffer (pH 8.5) containing 0.25M NaCl. Lipase activity was checked for each fraction and protein bands were also analyzed by using SDS–PAGE. The desired fractions were then pooled and protein contents were determined using Bradford assay (Bradford, 1976).

Lipase assay

Lipase activity was assayed calorimetrically per Kwon & Rhee (1986). The culture filtrate (one mL) was shaken with 2.5 mL of olive oil (70% oleate residues) emulsion (1:1, v/v) and 20 μL of 0.02M CaCl2 in a water bath shaker at an agitation rate of 200 rpm. The emulsion was prepared by mixing together an equal volume of olive oil (Bertolli, Pescaglia, Italy) and 50 mM phosphate buffer with a magnetic stirrer for 10 min. The reaction mixture was shaken for 30 min at 50 °C. The enzyme reaction was stopped by adding 6N HCl (one mL) and isooctane (five mL), followed by mixing using vortex mixer for 30 s. The upper isooctane layer (four mL) containing the fatty acid was transferred to a test tube for analysis. Copper reagent (one mL) was added and again mixed with a vortex mixer for 30 s. The reagent was prepared by adjusting the 5% (w/v) solution of copper (II) acetate-1-hydrate to pH 6.1 with pyridine. The absorbance of the upper layer was read at 715 nm. Lipase activity was measured by measuring the amount of free fatty acids. One μmol of lipase activity was measured as the free fatty acids generated in 1 min.

SDS–PAGE analysis

SDS–PAGE (12% running gel, 6% stacking gel) was conducted according to the method of Laemmli (1970). A broad range of protein standard (MBI Fermentas, St. Leon-Rot, Fermentas, Germany) was used as a molecular mass marker. Protein concentrations were determined according to the Bradford method, with bovine serum albumin as standard (Bradford, 1976).

Results

pI value computation

The pI values of GST tag and matured T1 lipase were theoretically determined by using ExPASy tool at 6.10 and 6.02, respectively, hindering their separation through anion exchange chromatography. Therefore, histidine at position 215 (H215) and glycine at position 213 (G213) of the amino acid were replaced with arginine. The new pI value of GST tag after computing H215R and G213R is 6.53, higher than the native GST tag by 0.43.

Site-directed mutagenesis at two amino acids was successfully performed using designed mutagenic primers with melting temperature (Tm) of 78 °C. Both of the 39 base pairs mutagenic primers (forward and reverse) contained mutations and were annealed to the opposite strands of the pGEX4T1 plasmids. These mutagenic primers were designed by locating the desired mutation sequence in the middle of the primer with 15 bases of correct sequence on both sides. Sequencing results for the mutated GST tag were analyzed online at the SDSC Biology Workbench website (http://workbench.sdsc.edu). The two points of mutations were successfully examined and no other changes in the GST sequence were found.

Purifications

The construct of GSTm was purified using two purification steps which were affinity chromatography and anion exchange chromatography. Affinity chromatogram results showed that GSTm interacted with the ligand coupled to a Glutathione Sepharose 4 Fast Flow and the elution was performed with 10 mM reduced glutathione as competitive ligand with flow rate of one mL/min with a recovery of 82%. Abundance of proteins was detected at a flow through with a reading 3,000 mAU at 280 nm and no lipase activities were determined at flow-through fractions, demonstrating that GSTm were successfully bound to the resins after go through some of modification. Elution was carried out with flow rate of one mL/min by the gradient with 20 CV (0–100%). The purified fusion lipase was eluted at 32% gradient, and one peak was obtained (Fig. 2A). The fractions of eluted fusion GST-T1 lipase were collected and pooled, assayed, and subjected to SDS–PAGE analysis. Lipase activity and SDS–PAGE gave 138.42 U/mL and 63 kDa of fusion GST-T1 lipase molecular size, respectively (Fig. 2B).

Figure 2 Chromatogram profile for affinity chromatography and SDS–PAGE analysis on eluted proteins.

(A) Chromatography result of affinity chromatography (AKTA Explorer) for GSTm-T1 lipase; a single peak were obtained. (B) SDS–PAGE analysis results for the purification of fusion GST-T1 lipase. Lane M, standard protein marker; Lane 1, crude enzymes; Lane 2–7 (Fraction 19–24), purified fusion GST-T1 lipase (66 kDa).

The separation of T1 lipase between modified GSTm and unmodified GST tag were examined. Digested GSTm and GST tag from T1 lipases was subjected to anion exchange chromatography with flow rate 0.5 mL/min. Theoretically, GSTm and T1 lipase mixtures can be separated through anion exchanger as their pI value have been differentiated (6.53 and 6.02, respectively) and raised by 0.43, otherwise not for unmodified GST tag as their pI values were close. The chromatogram result for native GST in Fig. 3A has proven that GST and T1 lipase were not well separated as two redundant peaks were obtained. SDS–PAGE analysis results revealed that T1 lipase was eluted with GST tags in all fractions, starting from Peak 1 followed by Peak 2, while Peak 3 represented pure GST tag (23 kDa). The fractions that contained partially purified T1 lipase at Peak 1 were pooled and the yield of mature T1 lipase obtained was only 11% of recovery.

Figure 3 Chromatogram profile of ion exchange chromatography for GST-T1 lipase and SDS–PAGE analysis on eluted proteins.

(A) Chromatography result of ion exchange chromatography (AKTA) for native GST-T1 lipase; overlapped peaks were obtained. (B) SDS–PAGE analysis results for the purification of a native T1 lipase. Lane M, standard protein marker; Lane 1–9, GST tag and T1 lipase eluted together at peak 1 (Fraction 24–32); Lane 10–12, GST tag and T1 lipase eluted together at peak 2 (Fraction 39–41); Lane 13 and 14: GST tag (Fraction 34 and 35).

However, the chromatogram results for GSTm-T1 lipase in Fig. 4A shows that three separated peaks were successfully obtained and peak 1 was revealed by SDS–PAGE as purified T1 lipase with 95.89 U/mL of pooled lipase activity SDS–PAGE result in Fig. 4B it is evident that single band of purified T1 lipase was successfully obtained as a purified enzymes, while the two bands were obtained for all fractions eluted at peak 2 and peak 3 were purely GST tag with size 23 kDa.

Figure 4 Chromatogram profile of ion exchange chromatography for GSTm-T1 lipase and SDS–PAGE analysis for eluted proteins.

(A) Chromatography result of ion exchange chromatography (AKTA) for mutated GST-T1 lipase; three separated peaks were obtained. (B) SDS–PAGE analysis results for the purification T1 lipase from new construct (affinity chromatography and ion exchange chromatography). Lane M, standard protein marker; Lane 1, crude cell lysate; Lane 2, purified fusion lipase (66 kDa); Lane 3, fusion lipase after thrombin cleavage at 16 °C; Lane 4, pooled purified matured T1 lipase at peak 1 (43 kDa); Lane 5, some of T1 lipase and GST tag eluted in the same fractions at peak 2; Lane 6, GST tag at peak 3 (23 kDa).

From IEX results we noticed that, experimentally, T1 lipase was eluted earlier instead of prediction that GST tag should eluted earlier as the pI value of GST tag is higher compared to T1 lipase. Even though the prediction is not same as in the experiment, the GST tag and the T1 lipase were separated after a pI value modification. The purity of T1 lipase pooled fractions was given in Table 1. The final yield of mature T1 lipase was 33%, with the later having a specific activity of 6,849 U/mg.

Table 1 Purification of T1 lipase from a newly constructed GSTm-T1 lipase from Escherichia coli BL21(DE3)pLysS as an expression system.

Purification	Volume (mL)	Protein content (mg/mL)	Activity (U/mL)	Total activity (U)	Specific activity (U/mg)	Yield (%)	Fold	
Crude	20	0.56	85.9	1,718	153.4	100	1	
Affinity	10.25	0.08	138.42	1,418	1,730.25	82	11	
IEX	5.9	0.014	95.89	565	6,849.28	33	44	

The purified T1 lipase obtained using new construct of GSTm-T1 lipase was compared to the previous traditional method conducted according to Aris et al. (2013). From comparison table in Table 2, the final recovery yield of traditional method was 19% which is lower than purification obtained using new construct of GSTm-T1 lipase which is 33%. Furthermore, the improvement of purification fold was observed using GSTm-T1 lipase with 44-fold compared to traditional method with only eightfold. The higher fold in purification profile explains that the degree of purity enzyme is higher. Furthermore, the affinity 2 method used in the traditional method was also costly and time consuming where three different types of resins were used as an attachable column and slower flow rate used at 0.25 mL/min. Consequently about 21 h time were required to complete the purification process. However, using new construct of GSTm-T1 lipase, only 5 h was taken to complete the overall purification process.

Table 2 Purification yield (%) and fold obtaining from previous purification strategies of T1 lipase.

Strategy	Resins	Time (h)	Yield (%)	Fold	
GST-T1 lipase	Affinity 1: Glutathione Sepharose FF	21	19	8	
Affinity 2: Glutathione Sepharose HP Hi Trap Glutathione FF Hi Trap Benzamidine	
Ion exchange chromatographyQ Sepharose HP	
Construct 1: GSTm-T1 lipase	Affinity 1: Glutathione Sepharose FF	5	33	44	
Ion exchange chromatography Q Sepharose FF	

Discussion

Rational design of GST pI value

Emerging an efficient purification strategy by some of trials error is time consuming. Therefore, the modelling of biochemical properties by rational design is a promising approach to allow focused experimental testing (Trodler, Rolf Schmid & Pleiss, 2008). The major finding of this work is successful purification by enlarging the differences of the pI values between the GST fusion tag and the matured lipase as obtained by site-directed mutagenesis, so that the two can be cleanly separated in anion exchange step. The theoretical pI value of proteins can be calculated and can be used as a reference for choosing proper conditions for purification (Chern et al., 2009). We computed the GST tag and T1 lipase pI value based on protein amino acid compositions in the sequence data. However, due to the close pI values of GST tag and T1 lipase by differences of 0.08, amino acid levels were changed to increase the pI value. Amino acid modification was only conducted on GST tag thus conformational structure of T1 lipase could be retained.

The GST tag expressed from the pGEX4T1 vectors is from Schistosoma japonica (Sj-GST) and it contains 218 amino acids. H215 and G213 were chosen as they located at the last eight amino acids in the Sj-GST domain (residues 211–218, GGGDHPPK) which not present in other mammalian GST structures and are unique for Sj-GST (Tigue, Williams & Tainer, 1995). It forms an open hairpin loop. The unique sequence of GGG at the bottom of the loop just behaves as a better linker to provide different proteins to its C terminus and does not affect the folding of GST tag. The linker regions after the open hairpin loop have different conformations for different sequences and really flexible for any amino acids changes. In other cases the substitution of Sj-GST Glu26 for His increases its metal binding affinity and allows for the efficient purification of recombinant proteins using immobilized Ni affinity chromatography (Han et al., 2010). The most relevant amino acids for changing the pI value of proteins is arginine (R) as pKa arginine’s side chain (12.50) is higher compared to other six ionizable residues. It has been commonly reported that substitution of amino acid to arginine often leads to enhancement and lowers the pI value of proteins (Zocchi et al., 2003).

Purification of T1 lipase

Many separation techniques can be used in order to separate the protein mixture. FPLC (AKTA Explorer) in combination with IEX is the most widely used method for industrial scaled-up purposes, as it can be used in either positive or negative capture modes. Affinity chromatography is frequently used as a first step for most protein purification studies. A previous purification strategy have been implemented using two steps of affinity chromatography (Leow et al., 2007) followed by IEX (Aris et al., 2014). Second affinity chromatography with attachable three columns (GSTrap FF, Benzamidine, and GSTrap HP) is not relevant to use in large scale. Therefore, the elimination of second affinity chromatography step in this study greatly enhanced direct separation of GST tag and T1 lipase using IEX.

The prediction of GST tag to be separated through IEX after pI value modification was experimentally confirmed by a Fig. 4A. Three separated peaks were obtained and the SDS–PAGE profile revealed that all fractions from Peak 1 are to be purified T1 lipase with molecular weight of 43 kDa (Fig. 4B). Nevertheless, an unmodified GST tag was unable to be separated on IEX, as redundant peaks were observed in Fig. 3. The purity of these pooled fractions is as given in Table 1.

Theoretically, pI value prediction shows an earlier elution of GST tag over matured lipase, but in this case matured lipase came out first, followed by GST tag. In reality, simple pI computation does not take the 3D conformation into account, so that the effects of intramolecular interactions between amino acids are ignored. If referred to a random coil, the pI value only depends on a protein amino acid composition and can thus be computed from analytical or sequence data. In contrast, in a native structure where interactions among amino acids stretches can occur, the experimental pI may be subtly altered. Measuring a pI value with precision is not an easy task. A pI is a physio-chemical parameter and its precise assessment might be quite precious in establishing the identity of a protein but a number of experimental parameters might alter its value. One way to inquire is to determine the pI values by native isoelectric focusing gel analysis, which could be compared with the denatured isoelectric focusing gel analysis. The theoretical pI values should be close to the results of denatured isoelectric focusing gel analysis, while the results from chromatography should be consistent with the results of native isoelectric focusing gel analysis.

The purity of the GSTm construct was compared to previous purification strategies (Table 2). From the previous traditional method, the final yield of mature T1 lipase after a two-step affinity and IEX was only 19%. A higher yield recovery of purified T1 lipase was obtained using this new purifying protocol at 33% recovery, thus skipping the affinity 2 steps, which are costly and time consuming when using three attached columns (GSTFF, GSTHP, and Benzamidine) at a lower flowrate of 0.25 mL/min. Using the newly proposed strategy in this study, the purification steps were successfully reduced from three steps to two steps. The final yield obtained in this strategy is 33% with high purity at the 44 purification fold, along with an acceptable yield and high fold obtained, indicating a high degree of T1 lipase purity. The higher fold of purification is due to high resolution and high loading capacity of IEX resins; therefore, the purification strategy of T1 lipase was successfully enhanced by using molecular modifications (Rathore & Vekayudhan, 2004).

Conclusions

This study has demonstrated that, via molecular modification, new separation and purification techniques were developed. The new constructs of pGEX4T1/GSTm-T1 with two purification steps were developed by replacing affinity 2 with IEX. The construct was developed by a rational design where double point mutations at two locations of the GST tag downstream region by raised the pI value of GST tag. The incensement of the pI value has improved the resolving between the GST tag and the T1 lipase peak through anion exchange chromatography. Nevertheless, comparisons have been made to Aris et al. (2013) purification strategy on time, yield and fold. The findings thus indicate that GSTm-T1 lipase appeared better with a 44-fold purification in 5 h of processing time than the traditional method of 21 h needed for completing the purification process. Therefore, this construct has strong potential for use in scaling up purification of T1 lipase, as it saved time while allowing for high levels of purification.

Supplemental Information

Supplemental Information 1 Theoretical pI value of T1 lipase and GST tag before and after pI value was computed at Expasy tool website.

(a) Theoretical pI value of T1 lipase. (b) Theoretical pI value of GST tag. (c) Theoretical pI value of GST tag after replacing two points of GST sequences. Glycine at position 213 replaced with Arginine (G213R) and Histidine at position 215 replaced with Arginine (H215R).

Click here for additional data file.

Supplemental Information 2 Sequence alignment result between GST and mutated GST (Mut_C2_MutSeq_Rev_Reverse-C).

Two points of mutation were successfully created at position 213-GGC (Glycine) changed to AGA (Arginine) and position 215-CAT (Histidine) changed to AGA (Arginine).

Click here for additional data file.

Supplemental Information 3 Graph of lipase activity for fractions during successful separation of GSTm tag with T1 lipase through ion exchange chromatography.

Click here for additional data file.

Additional Information and Declarations

Competing Interests

Author Contributions

Data Availability

The authors declare that they have no competing interests.

Che Haznie Ayu Che Hussian conceived and designed the experiments, performed the experiments, analyzed the data, contributed reagents/materials/analysis tools, prepared figures and/or tables, authored or reviewed drafts of the paper, approved the final draft.

Raja Noor Zaliha Raja Abd. Rahman conceived and designed the experiments, analyzed the data, contributed reagents/materials/analysis tools, authored or reviewed drafts of the paper, approved the final draft.

Adam Leow Thean Chor conceived and designed the experiments, analyzed the data, authored or reviewed drafts of the paper, approved the final draft.

Abu Bakar Salleh conceived and designed the experiments, analyzed the data, authored or reviewed drafts of the paper, approved the final draft.

Mohd Shukuri Mohamad Ali conceived and designed the experiments, analyzed the data, authored or reviewed drafts of the paper, approved the final draft.

The following information was supplied regarding data availability:

Raw data is available in the Supplemental File.

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
