# Peer review of "Enhancement of a protocol purifying T1 lipase through molecular approach"

_PeerJ, doi:10.7717/peerj.5833_

## Round 0.1 · original submission · Minor Revisions

Please address all the queries of both reviewers and amend your manuscript accordingly.

Reviewer 1 ·

Basic reporting

The authors used cleat language to describe their design rationales. Figures are well labeled and described, the main text is well referenced.

Experimental design

Methods are described in detail.

Validity of the findings

The findings are valid with robust data presentation.

Additional comments

The purification of GST-tagged T1 lipases suffered from low yield and cumbersome 3-step purification steps. Using rational design, the authors changed the PI value of the GST tag and shortened the purification to two steps, thereby increasing the overall yield quite significantly.

This is a very nice showcase of the power of rational design to improve protein engineering, and fits well with the theme of this journal. I highly support its publication if the following minor concerns can be addressed satisfactorily:

• Line 219, the subheading should be changed to be more informative.
• Figures 1 and 2 are not significant enough to be in the main text, I recommend moving to the supplements.
• Supplemental figures are missing figure captions. I recommend moving supp. Fig. 1 to the main text since it very nicely shows the overall workflow of this paper.
• Figures 3-5, the gel images should show better contrast for better readability. Lines with different colors should be labeled in the figure legends.
• On a similar note, font sizes on the X- and Y- axis in the panel (a) of Figures 3-5 should be increased.

Reviewer 2 ·

Basic reporting

I think the figure displaying in the paper is not so satisfying, some figures have occupied too much space while providing limited information. Like Figure 2, which I think can be moved to the supplementary data.

Experimental design

No comment

Validity of the findings

No comment.

Additional comments

This manuscript depicted a new protocol for T1 Lipase purification by mutating the GST tag. The English language used throughout the context is clear and professional. The experiments are well designed and right conclusions are obtained. The idea of changing the PI value of GST tag to improve the purifying process will provide new information for this field. However, I have some concerns about this paper.
1. Since the purification of T1 Lipase is a well-established method, this paper should put more emphasis on the comparison between the traditional method and the modified one. For example, in the result part, the difference between two strategies should be discussed in detail. And in table 2, why the data of GST-T1 Lipase was adopted from a published paper instead of your own data? I think it will be more persuasive to use your own data.
2. As far as I know, peptides can be separated using RP-HPLC based on their hydrophobic character. If this is the case, T1 Lipase will still be able to get purified from GST without mutating the tag. What do you think is the advantage of IEC over RP-HPLC for Lipase purification?
3. In Figure3-4, there are multiple lanes for the same sample, what’s the difference between them? Parallel samples? It should be stated clearly in the figure legend.
4. In Figure4, though there was some GST eluted together with T1 Lipase, the yield of T1 Lipase still looks pretty high; the yield should be stated with WT GST.
5. Quality of gel image is low in Figure3 and Figure4, the ladder is smear.
6. Figure 2 can be listed in supplementary data.

---

## Round 0.2 · accepted · Accept

Thank you for addressing all critical points raised by the reviewers and for corresponding amendments of the manuscript. The revised version is acceptable now.

#